# Low-Power Driving Waveform Design for Improving the Display Effect of Electrophoretic Electronic Paper

**DOI:** 10.3390/mi15091076

**Published:** 2024-08-26

**Authors:** Shanling Lin, Jianhao Zhang, Jia Wei, Xinxin Xie, Shanhong Lv, Ting Mei, Tingyu Wang, Bipeng Cai, Wenjie Mao, Tailiang Guo, Jianpu Lin, Zhixian Lin

**Affiliations:** 1School of Advanced Manufacturing, Fuzhou University, Quanzhou 362251, China; sllin@fzu.edu.cn (S.L.); 228527161@fzu.edu.cn (J.Z.); 238527297@fzu.edu.cn (J.W.); 218527016@fzu.edu.cn (X.X.); shanhglv2022@fzu.edu.cn (S.L.); tingyuw0@gmail.com (T.W.); 228527283@fzu.edu.cn (W.M.); gtl@fzu.edu.cn (T.G.); 2National Local United Engineering Lab of Flat Panel Display Technology, Fuzhou University, Fuzhou 350116, China; 221110007@fzu.edu.cn (T.M.); 221120001@fzu.edu.cn (B.C.); 3College of Physics and Telecommunication Engineering, Fuzhou University, Fuzhou 350116, China

**Keywords:** electrophoretic electronic paper, refresh power consumption, driving waveforms, low power consumption, flicker, ghosting

## Abstract

To address the high power consumption associated with image refresh operations in EPDs, this paper proposes a low-power driving waveform that reduces the refresh power of EPDs by lowering the system’s peak power. Compared to traditional waveforms, this waveform first activates the particles before erasing them, thus reducing voltage polarity changes. Additionally, it introduces a specific duration of 0 V voltage during the activation phase based on the physical characteristics of the electrophoretic particles to reduce the voltage span. Finally, a particular duration of 0 V voltage is introduced during the erasure phase to minimize the voltage span while ensuring the stability and consistency of the reference gray scale. The experimental results demonstrate that, in standard power tests, the new driving waveform reduces the power fluctuation value by 1.33% and the energy fluctuation value by 37.24% compared to the traditional driving waveform. This reduction in refresh power also mitigates screen flicker and ghosting phenomena.

## 1. Introduction

Electrophoretic electronic paper (EPD) is a display technology that simulates the appearance of paper. It has advantages such as low power consumption, reflectivity, biostability, etc. [1]. EPDs are widely used in various fields, including e-books, e-labels, electronic billboards, and electronic license plates, providing convenience and comfort in daily life and work. With continuous technological advancements, EPDs are also evolving towards color displays, video capabilities, and flexibility [2].

The display of EPDs relies on the movement of particles driven by different timing voltages. The magnitude of these voltages and the duration of the driving time determine the position of the particles, thus the gray scale of the electronic paper (e-paper) display [3]. This timing voltage composition is known as the driving waveform of EPDs. During the driving process, the size and duration of the applied voltage in the driving waveform affect the position of the electrophoretic particles in the microcapsules. The power consumed to drive these particles varies depending on their positions and the driving waveforms used. Additionally, due to the viscous nature of electrophoretic particles, considerable time is often required to align the particles to achieve the desired gray scale, sometimes taking hundreds of milliseconds or even a full second. The traditional driving waveform is typically divided into three stages: the elimination of the original image, particle activation, and writing a new image. The elimination stage stabilizes the EPD screen, usually turning it white or black to reduce residual shadows. The activation phase increases particle activity by repeatedly driving them between the optical extremes (black and white), reducing the sticking effect and making it easier to write new data. Finally, the writing stage drives the e-paper to display new shades of gray.

Due to its inherent bistable characteristics similar to memory behavior, EPDs maintain the image display for an extended period without power and consume power only when the display needs to be refreshed or updated. In practice, the power consumption of EPDs arises from the protocols during data transmission, the switching or refreshing of the TFT source lines, and the driving waveforms [4]. Therefore, optimizing power consumption mainly focuses on these aspects. Li W et al. [5] addressed the high power consumption of EWD displays by analyzing the influence of the driving waveform and designing a waveform with a rising gradient and a sawtooth pattern to reduce the power consumption. Qingyun Luo et al. [6] tackled the issue of increased power consumption in e-paper displays due to halftone technology by proposing a multi-objective optimization technique to simultaneously optimize the image quality and power consumption in color electrophoretic displays, achieving a Pareto optimum. Pitt et al. [4] analyzed the power consumption caused by capacitive losses due to switching the source lines of TFTs and proposed reducing the switching voltage to a lower power dissipation, although this approach increased the response time. JY Kim et al. [7] developed a new driving scheme to reduce power consumption by minimizing the number of driving data lines of TFTs and designing a new TFT panel structure. Cheng Wei et al. [8] proposed a voltage-driven waveform debugging method to reduce e-paper power consumption by staggering the positive and negative voltage segments of each color-developing particle at different times, thereby reducing the IC load and average power consumption during screen refreshes.

In summary, the power consumption of the driving waveform mainly arises from increasing the number of gray levels or the accuracy, or from particles flipping back and forth between two optical limit states. To further optimize the driving waveform and reduce the refresh power, this paper investigates the factors contributing to EPD power consumption, incorporates the characteristics of traditional driving waveforms, and proposes an optimized waveform that reduces both the refresh power and the phenomena of flickering and ghosting.

## 2. Principle

### 2.1. Principle of EPDs

Inside the EPDs, positively and negatively charged particles, along with colorless transparent organic solvents, are encapsulated in tiny capsules fixed in a transparent adhesive [9]. EPDs display an image by applying an electric field to the e-ink, causing the charged pigment particles to move through a nonpolar solvent under the influence of the electric field [10], as illustrated in Figure 1. Under the control of an external electric field, the particles move according to the principle of “homopolar repulsion and anisotropic attraction” to achieve the image display effect [11]. The microcapsules are embedded in a binder layer (binder 1), with the microcapsule walls separating the electronic ink. Pixel electrodes adhere to the microcapsule layer through another binder layer (binder 2). These pixel electrodes are connected to the thin-film transistors of active matrix EPDs, generating an electric field to drive the particles [12].

### 2.2. Factors That Generate Power Consumption

The power consumption of EPDs primarily originates from two aspects. The first is the refresh operation, which involves displaying a new image or text on the screen. This process requires the rearrangement of the electronic ink particles inside the EPD to form a new image or text, thereby consuming power. The second is the control circuit that drives the EPD display. This circuit is responsible for generating the electric field, processing image data, performing the refresh operation, and ensuring the stability of the display content, all of which contribute significantly to power consumption.

E-paper is bistable, meaning that the particles remain in their current position even without an electric field, allowing it to consume almost no power when the display is stationary. The refresh operation of EPDs involves moving the electronic ink particles to display a new image or text, which consumes electrical energy. Each pixel of an EPD consists of tiny capsules containing charged particles of various colors, and the speed and path of these particles are influenced by the strength and direction of the electric field. Therefore, the precise control of the driving waveform is crucial for the refresh operation.

Different driving waveforms need to be designed according to the photoelectric properties of the particles and the target gray scale, with each driving waveform representing a specific voltage timing. Different voltage timings result in varying power consumption, making the design of driving waveforms a critical factor in the refresh power of EPDs. Poorly designed driving waveforms can lead to excessive transient currents and increased power consumption.

During the driving waveform design process, a sudden change in voltage can cause rapid charging and discharging of the capacitor, generating transient current and leading to peak power.

The peak power satisfies the following relationship [13]:(1)$$P_{peak}=\frac{1}{2}C{\left(\Delta V\right)}^{2}\cdot f\cdot V_{source}$$
where *C* is the capacitance value, Δ*V* is the magnitude of the voltage variation, *f* is the switching frequency, and *V_source_* is the supply voltage.

From Equation (1), it can be seen that the magnitude of the voltage change will cause larger peak power when the other conditions remain unchanged, Therefore, in the design of the driving waveform, reducing the number of voltage polarity changes and the voltage amplitude can effectively lower the transient power. The reduction in peak power will reduce the overall refresh power of EPDs.

## 3. Experiment and Discussion

### 3.1. Experiment Platform

#### 3.1.1. Optoelectronic Performance Test Platform

In this study, a microcapsule e-paper (10.3 inches, with a screen resolution of 1680 × 2240, a driving voltage of ±15 V, and a driving frequency of 66.67 Hz) manufactured by BOE (Beijing, China) was used for the experiments. To avoid interference from external environmental factors on the measurement of EPD reflectivity, a darkroom test system was designed, as shown in Figure 2. A gray scale response time meter, FS-GRT, was placed directly in front of the EPD under testing. This setup accurately collected the photoelectric change data and calculated the gray scale response time and flicker data.

Based on the selection of the EPD’s driving chip, a drive board based on an FPGA was designed and developed for the EPD display. Additionally, the UNI-T UTP1310 switching regulator power supply was used to power the light source, providing a stable output voltage and current. This stable power supply effectively reduces any interference with the test results caused by power supply fluctuations.

#### 3.1.2. Power Test Platform

In this study, a power test system device was designed, as shown in Figure 3. A power analyzer, EKA1080M (Emkia, Shenzhen, China), was used to measure the total current of the drive board and EPD over time. These data were then used to calculate the refresh power of the EPD.

### 3.2. Optoelectronic Performance Test

The conventional EPD driving process is divided into three stages: erasure, activation, and display. To determine the appropriate driving time for each stage of the selected EPD, the photoelectric characteristics were studied. First, the EPD underwent multiple rounds of positive voltage refresh operations. The effect of different voltage durations on screen reflectivity at +15 V was measured using a gray scale response time meter, with the results shown in Figure 4a. This step aimed to determine the time required for all black particles to be effectively driven to the common electrical extremity under different voltage durations. The initial reflectance was approximately 0.75, decreasing nonlinearly with an increased positive voltage application time, stabilizing around 0.1 after 90 ms.

Next, the EPD was subjected to multiple rounds of negative voltage refresh operations. A gray scale response time meter measured the effects of different voltage durations on screen reflectivity at −15 V, with the results shown in Figure 4b. This step aimed to determine the time required for all the white particles to be effectively driven to the common electrical pole. The initial reflectivity was around 0.1, increasing nonlinearly with an increased negative voltage application time, stabilizing around 0.7 after 90 ms, though not reaching the maximum reflectivity of 0.75. Therefore, the drive-to-black and drive-to-white times in the activation stage of the driving waveform were set to 90 ms, sufficient to shift the EPD’s screen reflectivity between 0.1 and 0.7.

During the design of the EPD’s driving waveform, single-voltage driving can lead to significant gray scale loss. To avoid this, a multi-stage voltage combination strategy was used, with subframe-based driving adopted. Each subframe was 15 ms. When selecting the reference gray scale, a state with high stability must be chosen, typically white or black [14]. Generally, the white gray scale is used as the reference, and other gray scales are derived from it [15]. The experimental results showed that using white as the reference gray scale yielded a better display quality than using black. Thus, a negative voltage was used to drive the EPD to display the white reference gray scale, and a positive voltage was used for the target gray scale adjustment.

In this paper, four gray scales were used to verify the waveform’s validity: 0.7 (white), 0.5 (light gray), 0.3 (dark gray), and 0.1 (black).

### 3.3. Low-Power Driving Waveform Design

A traditional driving waveform structure is shown in Figure 5a, where the driving waveform of conventional EPDs generally includes an erase phase, an activation phase, and a write phase. According to the peak power formula *p* = *C*/2 × *f* × (Δ*V*)^2^ × *V_source_*, it can be found that traditional waveforms have more voltage reversals, so the peak power consumption is larger. TA1 = TD1 and TB1 = TC1 in Figure 5a to satisfy the DC balance principle.

For the low-power driving waveform, it is necessary to reduce the voltage span while ensuring the DC balance and display quality to avoid damage to the e-paper screen. The low-power driving waveform structure proposed in this paper is shown in Figure 5b. In this structure, the activation voltage is applied first, followed by the erasure voltage. This approach reduces the peak power by decreasing the number of voltage polarity changes compared to the traditional waveform. During the activation stage, the black and white particles move at different speeds under the same voltage due to their distinct characteristics, resulting in different times to reach their extreme optical states. Therefore, it is unnecessary to keep the driving times to black and white the same during the activation stage; it is sufficient to drive both particles to their extreme optical states. The difference between the two is TX, to which is applied a 0 V voltage, reducing the voltage span and lowering the peak power.

The erasure stage then follows, applying a voltage of TA1 duration to cycle the DC balance and erase the original image. A 0 V voltage voltage of TY - is then applied to allow the particles to reach a steady state on their own, forming a stable and consistent reference gray scale. This also reduces the voltage span and lowers the peak power.

In this paper, four gray levels are used for the waveform verification experiments. The waveforms for B-B, B-DG, B-LG, B-W, DG-W, LG-W, and W-W of the conventional driving waveforms are shown in Figure 6a. The distance between the two dotted lines in the figure represents a minimum time unit of 15 ms. Based on the test results of the optoelectronic characteristics, to ensure that the particles are fully activated while avoiding excessively long response times, each stage’s waveform length is designed to be longer than 75 ms. Thus, the first stage is a single-DC balancing stage with a length of 90 ms, the second stage is the activation stage with a length of 180 ms, and the third stage is the new image writing stage with a length of 90 ms.

A set of low-power driving waveforms for four gray levels (B-B, B-DG, B-LG, B-W, DG-W, LG-W, and W-W) is shown in Figure 6b. The activation stage is performed first, with a zero-setting voltage duration TX, which is optimized to 15 ms according to the response time and the movement speed of the black and white particles. Next, the erase stage is carried out, and the length of the new image writing stage is set to 180 ms. Depending on the previous gray state and the principle of cyclic DC balance, different lengths of erasing voltage are applied, with the zeroing voltage duration, TY, optimized to 15 ms to balance the response time and display effect.

### 3.4. Gray Scale Display Effect Comparison

In this paper, four gray scale driving waveforms are used to compare the effects of a gray scale display, as shown in Figure 7. Figure 7a shows the brightness values of four gray scales in cd/m^2^ under traditional waveform driving, while Figure 7b shows the brightness values under low-power waveform driving. From the brightness comparison, it can be seen that under the low-power driving waveform, the EPD display has a higher contrast, and the luminance differences between the gray scales are more uniform and reasonable. This results in a better display effect compared to the traditional driving waveform.

### 3.5. Power Test

Figure 8 shows the standard images used for e-paper refresh power tests according to the IEC 62679-3-2:2013 [16] and GB/T 43789.32-2024 [17] standards. Figure 8a displays a checkerboard pattern with 50% coverage, while Figure 8b shows its inverted checkerboard pattern. The power measured during the refresh process of EPDs from pattern a to pattern b is considered the refresh power of the EPDs; and patterns a and b should have the same contrast.

In this paper, we use the EKA1080M power analyzer to measure and calculate the refresh power of the EPD display, and the relationship between the current and time in the process of refreshing the image using the driving waveform is obtained by the power analyzer as shown in Figure 9:

According to the IEC 62679-3-2:2013 and GB/T 43789.32-2024 standards, the e-paper refresh power can be calculated by Formula (2):(2)$$W=\int_{0}^{t}VIdt$$
where *V* represents the voltage; *I* represents the current; and *W* represents the electrical energy.

Because the driving circuit of the e-paper is continuously providing power during the image refresh process, in order to better compare the change of power during the image refresh process, according to the definition of the power fluctuation value of a flat-panel TV in GB 24850-2020 “Flat-panel TVs and Set-top Boxes Energy Efficiency Limit Values and Energy Efficiency Levels” [18], the power fluctuation value of e-paper is determined by the absolute value of the difference between the static display power and the refresh power, divided by the static display power, which is *U_wt_*. The absolute value of the difference between the static display power and refresh power and the ratio of the static display power is defined as the e-paper power fluctuation value, symbolized as *U_pt_*, corresponding to the e-paper power fluctuation value of *U_wt_*. The power fluctuation value can be calculated by Formulas (3) and (4).

The power fluctuation value of the conventional driving waveform is as follows:(3)$$U_{pt}=\frac{P_{t}-P_{s}}{P_{s}}$$

The power fluctuation value of the low-power driving waveform is as follows:(4)$$U_{pl}=\frac{P_{l}-P_{s}}{P_{s}}$$

The energy fluctuation value of the conventional driving waveform is as follows:(5)$$U_{wt}=\frac{P_{t}\cdot t_{1}-P_{s}\cdot t_{1}}{P_{s}\cdot t_{1}}=U_{pt}$$

The energy fluctuation value of the low-power driving waveform is as follows:(6)$$U_{wl}=\frac{P_{l}\cdot t_{2}-P_{s}\cdot t_{1}}{P_{s}\cdot t_{1}}$$

The power fluctuation value reduction rate *R_p_* of the low-power waveform compared to the conventional waveform can be calculated by Equation (7):(7)$$R_{p}=U_{pt}-U_{pl}$$

The energy fluctuation value reduction rate *R_w_* of the low-power waveforms compared to the conventional waveforms can be calculated by using Equation (8):(8)$$R_{w}=U_{wt}-U_{wl}$$
where *U_pt_* represents the power fluctuation value of the traditional waveform; *U_pl_* represents the power fluctuation value of the low-power waveform; *U_wt_* represents the energy fluctuation value of the traditional waveform; *U_wl_* represents the energy fluctuation value of the low-power waveform; *P_t_* represents the refresh power of the traditional waveform; *P_l_* represents the refresh power of the low-power waveform; *P_s_* represents the static display power; *R_p_* represents the power fluctuation value reduction rate of the low-power waveform compared with the traditional waveform; *R_w_* represents the energy fluctuation value reduction rate of the low-power waveform compared with that of the traditional waveform; *t_1_* represents the traditional driving waveform refresh cycle; and *t_2_* represents the low-power driving waveform refresh cycle.

In order to avoid experimental chance, this paper performed 10 independent repetitive experiments, and the experimental measured data after the calculations are as shown in Table 1.

Table 1 demonstrates that the refresh power of the EPD is significantly higher than its static display power. Under the low-power driving waveform, the refresh power of the EPD is reduced compared to the traditional driving waveform, verifying the feasibility of this low-power driving approach. After the testing and calculations, we found that the power fluctuation value of the EPD with the low-power driving waveform is 1.33% lower than that with the traditional driving waveform, and the overall power consumption is reduced by 37.24%.

Additionally, this paper randomly selected nine pictures with different scene styles to perform the driving waveform power test, as shown in Figure 10. The refresh method involved transitioning from a white screen to each picture.

According to the above standard power test method, in order to avoid chance, 10 independent repetitions of the experiment were performed for each picture, and the average refresh power data of each picture was obtained as shown in Table 2:

The optimized value Δ represents the reduction in refresh power of the EPD under the low-power driving waveform compared to the traditional driving waveform. As shown in Table 2, while the low-power driving waveform optimizes the refresh power for all images, the optimization values vary across different images. Therefore, the average refresh power of the nine images is used for the final comparison in this paper. The negative power fluctuation value of the low-power driving waveform when refreshing, as seen in Figure 3 and Figure 6, indicates that the power consumed by refreshing these two images under the low-power driving waveform cycle is less than the power consumed by statically displaying the images under the traditional driving waveform cycle. The experimentally measured static display power of white images is 0.4114 W. According to Equations (3)–(8), the average power fluctuation value of the low-power driving waveform for the nine images under the low-power driving waveform is 6.75% lower, and the power fluctuation value is reduced by an average of 27.37% compared to the traditional driving waveform.

### 3.6. Display Effect Test

#### 3.6.1. Flicker Test

Figure 11 and Figure 12 show the process of image switching of EPDs under conventional waveform driving and low-power driving waveforms, respectively. Under conventional driving waveforms, the screen flickers four times, and, under low-power driving waveforms, the flickering of the EPD’s once-refreshed image is reduced to three times.

#### 3.6.2. Ghosting Test

Figure 13 shows the phenomenon of ghosting in EPDs under the traditional driving waveform and the low-power driving waveform of this paper. Under the traditional driving waveform, the ghosting of the screen is seriously retained, which seriously interferes with the clear display of the subsequent images. Under the low-power driving waveform proposed in this paper, the ghosting is basically eliminated, which significantly improves the clarity of the screen display and brings a more comfortable reading experience to users.

## 4. Conclusions

To address the high power consumption issue of electrophoretic electronic paper during an image refresh, a low-power driving waveform that reduces the refresh power of EPDs by lowering the peak power of the system was proposed by analyzing the characteristics of EPDs. The final experimental results show that the low-power driving waveform reduces the power fluctuation value by 1.33% and the energy fluctuation value by 37.24% compared to the traditional driving waveform. This reduces the refresh power of the electrophoretic e-paper while also decreasing the screen flicker and ghosting images.

## Figures and Tables

**Figure 1 micromachines-15-01076-f001:**
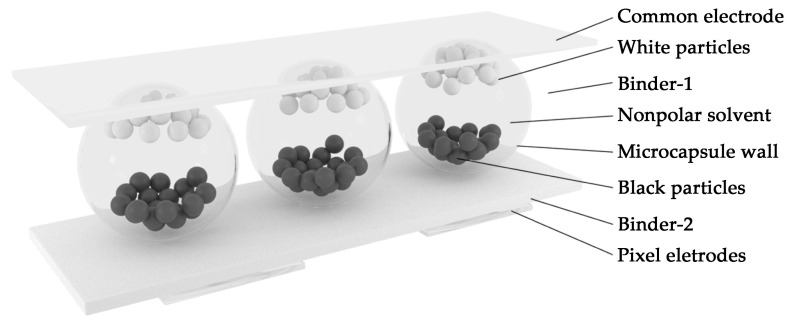
Schematic diagram of the microcapsule EPD structure. At the top is the common electrode, followed by the microcapsule layer. The microcapsules are suspended in binder 1. The microcapsules contain a nonpolar solvent in which black and white particles are dispersed. At the bottom is the pixel electrode, which adheres to the microcapsule layer of binder 2.

**Figure 2 micromachines-15-01076-f002:**
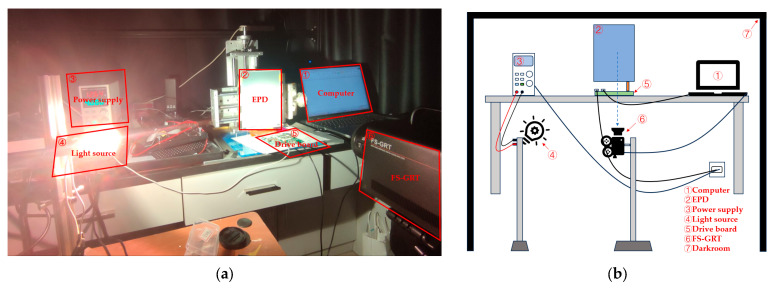
The EPD optoelectronic performance test platform, measuring the relationship between the EPD’s reflectivity and response time. (**a**) Physical figure of the optoelectronic performance test platform; and (**b**) model figure of the optoelectronic performance test platform.

**Figure 3 micromachines-15-01076-f003:**
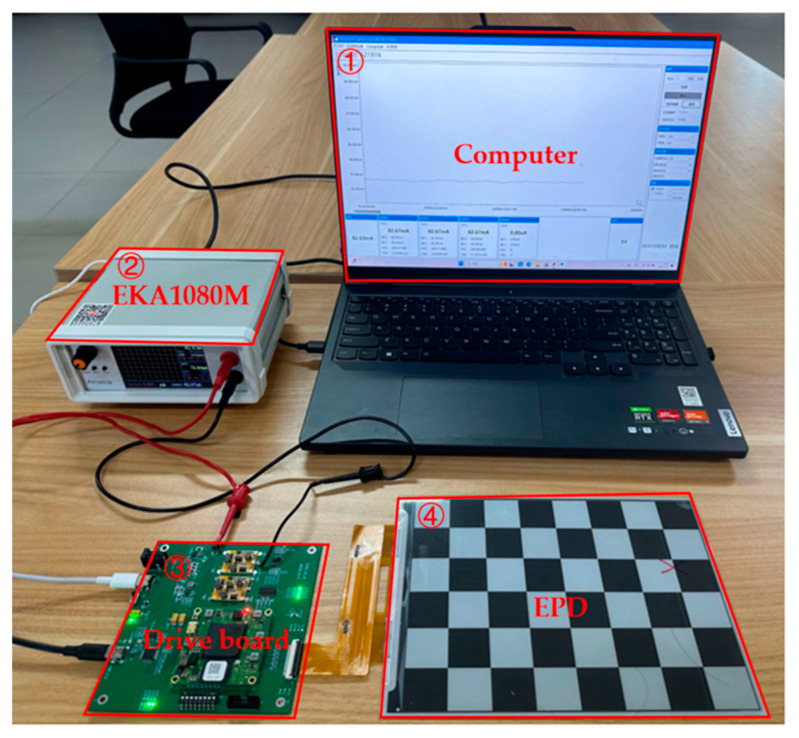
The EPD’s power consumption test bench for the measurement of the EPD’s current versus time: (1) computer; (2) EKA1080M; (3) drive board; and (4) EPD.

**Figure 4 micromachines-15-01076-f004:**
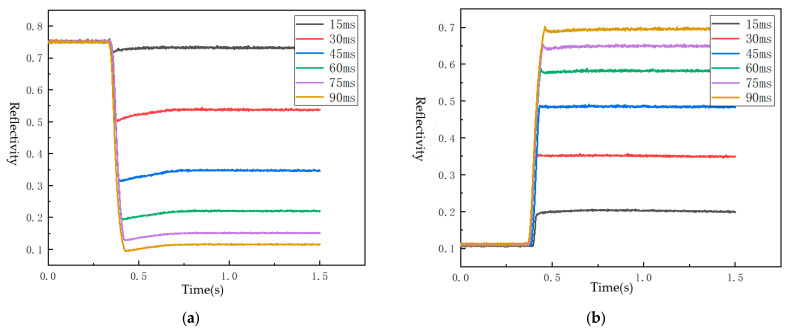
Relationship between voltage and e-paper reflectivity for different driving durations: (**a**) change in e-paper reflectivity at +15 V for the varying driving duration; and (**b**) change in e-paper reflectivity at −15 V for the varying driving duration.

**Figure 5 micromachines-15-01076-f005:**
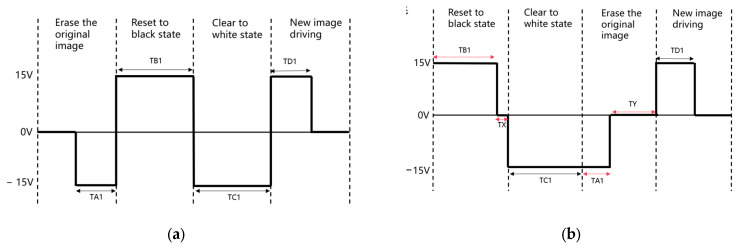
Comparison of driving waveform structures: (**a**) conventional driving waveform structure; and (**b**) the low-power driving waveform structure of this paper.

**Figure 6 micromachines-15-01076-f006:**
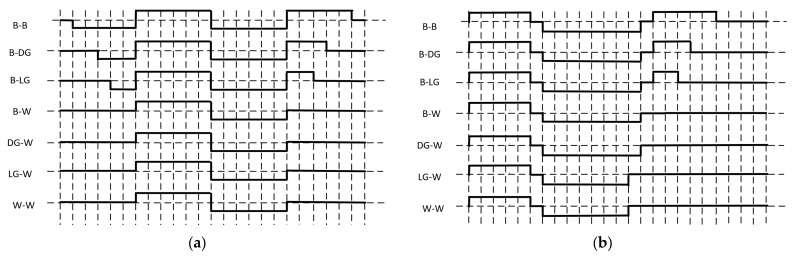
Comparison of driving waveform examples: (**a**) conventional driving waveform example; and (**b**) low-power driving waveform example.

**Figure 7 micromachines-15-01076-f007:**
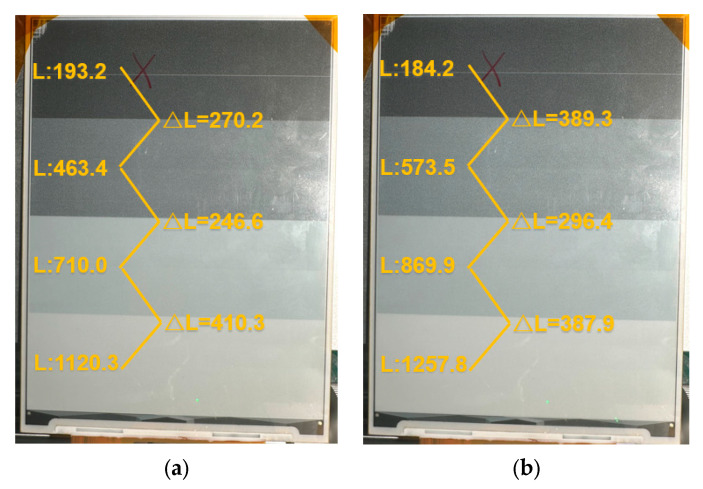
Comparison of four gray scale brightnesses: (**a**) four gray scale brightnesses with conventional driving waveforms; and (**b**) four gray scale brightnesses with low-power driving waveforms.

**Figure 8 micromachines-15-01076-f008:**
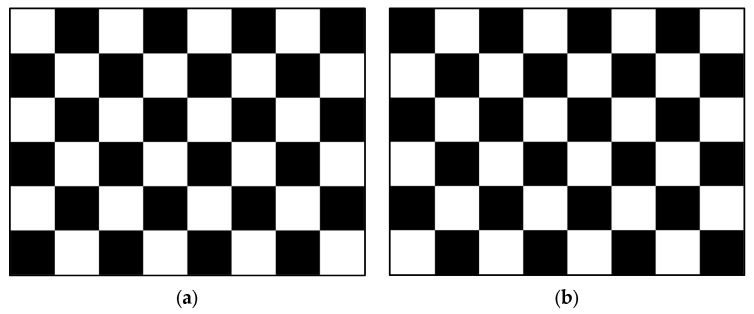
Standard refresh power test plot: (**a**) checkerboard pattern with 50% coverage; and (**b**) checkerboard pattern inverted from pattern a.

**Figure 9 micromachines-15-01076-f009:**
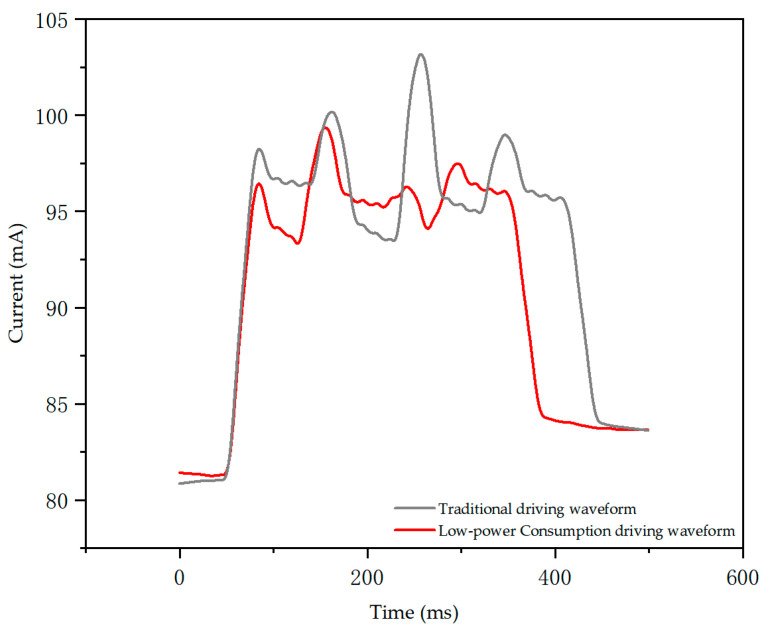
Current time variation graph during image refresh.

**Figure 10 micromachines-15-01076-f010:**
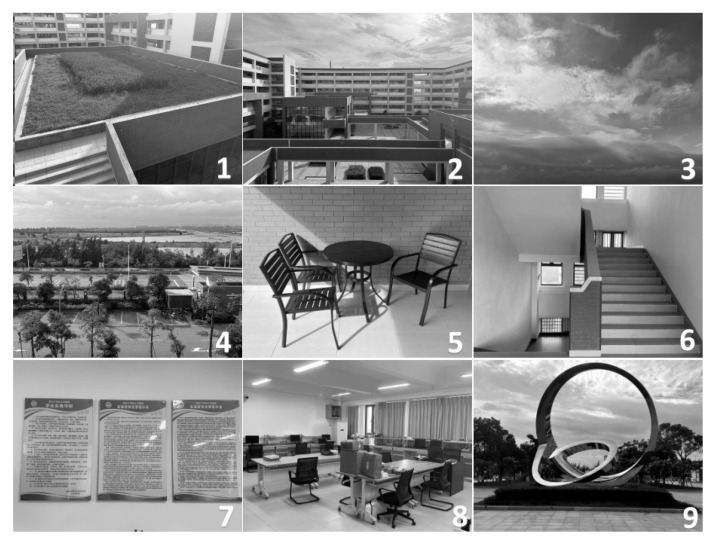
Power consumption test with random black and white pictures.

**Figure 11 micromachines-15-01076-f011:**
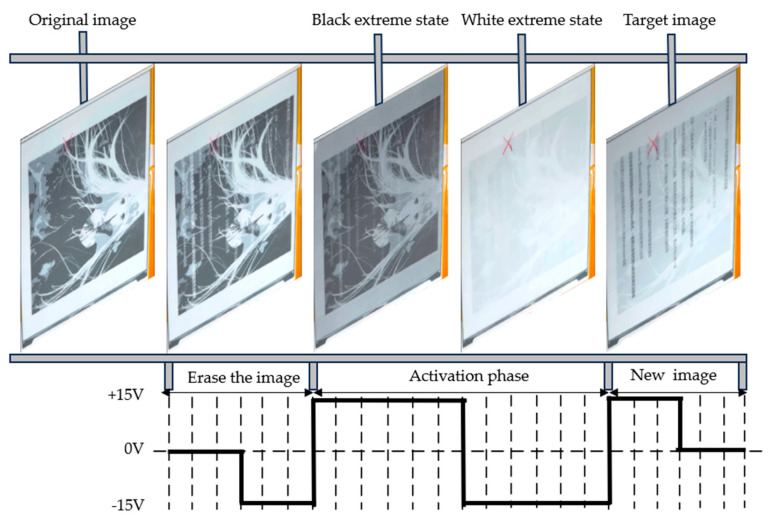
Switching process of electrophoretic display under conventional driving waveform.

**Figure 12 micromachines-15-01076-f012:**
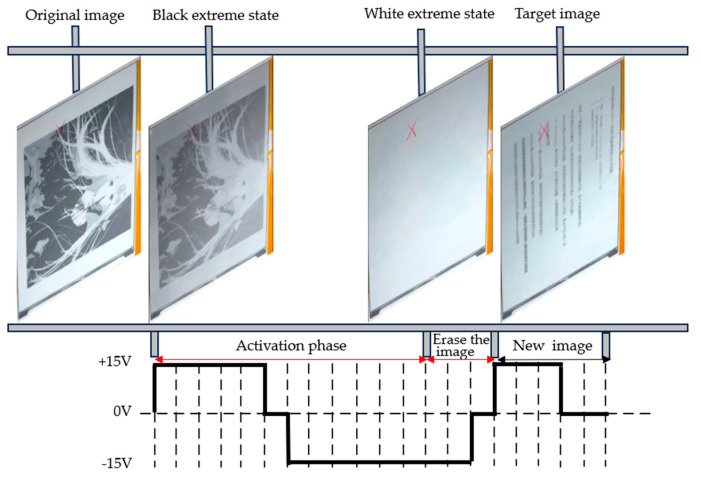
Electrophoretic display switching process under the low-power driving waveform of this paper.

**Figure 13 micromachines-15-01076-f013:**
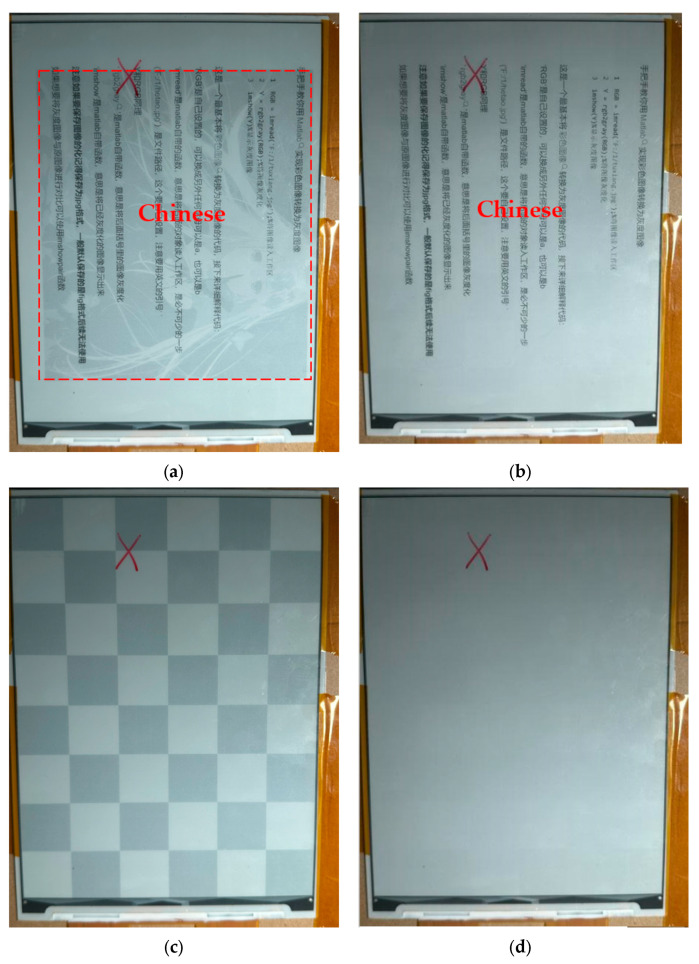
Ghosting phenomenon: (**a**) refresh from image of a girl to text under conventional driving waveform; (**b**) refresh from image of a girl to text under low-power driving waveform; (**c**) refresh from checkerboard to white gray scale under conventional driving waveform; and (**d**) refresh from checkerboard to white gray scale under low-power driving waveform.

**Table 1 micromachines-15-01076-t001:** Refresh power and static display power values of EPDs with different driving waveforms.

	Refresh Power (W)	Static Display Power (W)
Conventional Driving Waveforms	0.4852	0.4055
Low-power Driving Waveforms	0.4798	

**Table 2 micromachines-15-01076-t002:** Average power value of 10 refreshes of nine images under different driving waveforms.

Experimental Scenario	*P_t_* (*W*)	*P_l_* (*W*)	Optimal Value Δ*P* (*W*)	*U_pt_*/*U_wt_* (%)	*U_pl_* (%)	*U_wl_* (%)
1	0.5652	0.5256	0.0396	37.38	27.76	6.47
2	0.5323	0.5065	0.0258	29.39	23.12	2.60
3	0.4964	0.4858	0.0106	20.66	18.08	−1.6
4	0.5797	0.5333	0.0464	40.91	29.63	8.03
5	0.5308	0.5053	0.0255	29.02	22.82	2.35
6	0.5068	0.4916	0.0152	23.19	19.49	−0.42
7	0.5689	0.5272	0.0417	38.28	28.15	6.79
8	0.5349	0.5077	0.0272	30.02	23.41	2.84
9	0.5142	0.4963	0.0179	24.99	20.64	0.53
average	0.5366	0.5088	0.0278	30.43	23.68	3.06

## Data Availability

Data are contained within the article.

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
