# Peer review of "Low-Power Driving Waveform Design for Improving the Display Effect of Electrophoretic Electronic Paper"

_micromachines, 2024, doi:10.3390/mi15091076_

Round 1

Reviewer 1 Report

Comments and Suggestions for Authors

This paper investigates the factors contributing to EPDs power consumption and proposes a low-power driving waveform that reduces the refresh power of EPDs by lowering the system's peak power. Compared to the traditional driving waveform, this waveform not only reduces the refresh power, but also mitigates screen flicker and ghosting phenomena. So, this work has good practical value. However, there are some issues that need to be revised in the manuscript. If they can be carefully revised and improved, I recommend publishing this manuscript.

1. This paper has some formatting issues,the subheading numbers in the paper are duplicated and inconsistently labeled, with two sections labeled 3.2. The last sentence of the second paragraph under the second 3.2 section, which fails to convey the content clearly and may confuse readers. Please revise and thoroughly proofread the entire text.

2. There are spelling errors in the article, such as "ghosting" being misspelled as "gghosting" and "rghosting" in section 3.5.2. Please check the spelling throughout the paper and correct any mistakes.

3.The article mainly discusses issues related to power and energy. The formula W=Pt in equation (3) is a commonly used equation that most researchers in the field would understand, so it may not be necessary to explicitly list it.

 4.Why not directly compare the refresh power under different driving waveforms in the paper, but define the power fluctuation value and energy fluctuation value for comparison?

Author Response

Comment 1:This paper has some formatting issues,the subheading numbers in the paper are duplicated and inconsistently labeled, with two sections labeled 3.2. The last sentence of the second paragraph under the second 3.2 section, which fails to convey the content clearly and may confuse readers. Please revise and thoroughly proofread the entire text.

Response 1:Thank you for pointing this out. We agree with this comment. Therefore,We have revised the serial number of section 3.2 and the following sections and corrected the grammatical errors in the second section 3.2.

Comment 2:There are spelling errors in the article, such as "ghosting" being misspelled as "gghosting" and "rghosting" in section 3.5.2. Please check the spelling throughout the paper and correct any mistakes.

Response 2:Thank you for pointing this out. We agree with this comment. Therefore,we have corrected the spelling of the wrong words in section 3.5.2. and checked the spelling of the full-text words.

Comment 3:The article mainly discusses issues related to power and energy. The formula W=Pt in equation (3) is a commonly used equation that most researchers in the field would understand, so it may not be necessary to explicitly list it.

Response 3:Thank you for pointing this out. We agree with this comment. Therefor,we have deleted the equation (3).

Comment 4:Why not directly compare the refresh power under different driving waveforms in the paper, but define the power fluctuation value and energy fluctuation value for comparison?

Response 4:Because the driving circuit of electronic paper continues to provide power during the image refreshing process, defining power fluctuation value and electric energy fluctuation value can better compare the changes of power and electric energy during the image refreshing process.

Reviewer 2 Report

Comments and Suggestions for Authors

In this paper, the authors have presented "Driving Waveform Design for Improving the Display Effect of Electrophoretic Electronic Paper". I will give just one comment. 

First, I would like to confirm that the comparison of the power consumption is only shown in Table 1 ? In Table 1, the difference in the power consumption is not so different between the conventional and proposed drivings. So, it is a little unnatural to use and enhance "Low- power" in the title.

Author Response

Comment 1:First, I would like to confirm that the comparison of the power consumption is only shown in Table 1 ? In Table 1, the difference in the power consumption is not so different between the conventional and proposed drivings. So, it is a little unnatural to use and enhance "Low- power" in the title.

Response 1:Thank you for your question. The comparison of power consumption is not only shown in Table 1 but also in Table 2. Table 1 shows the comparison of standard checkerboard refresh power under the internationally regulated electrophoretic electronic paper display power testing method, while Table 2 shows the comparison of refresh power for various types of images measured using this method. The data in Table 2 is more universal. The difference in refresh power between traditional and proposed drivers in Table 1 is not significant because the driving circuit continues to provide power during the image refresh process. The power fluctuation caused by the static display of the image to refresh the image in the working state of the driving circuit is relatively small compared to the static display power. Therefore, simply comparing the refresh power cannot reflect the problem well. Therefore, we defined power fluctuation values and energy fluctuation values to assist in comparing the power consumption reduction effect of the proposed waveform relative to the traditional waveform.